# Abdominal obesity and association with sociodemographic, behavioral and clinical data in climacteric women assisted in primary care

Wiviane da Costa Pimenta[1], Josiane Santos Brant Rocha[1,2], Antônio Prates Caldeira[1,2], Daniela Araújo Veloso Popoff[1,2], Viviane Maia Santos[2], Joyce Elen Murça de Souza[1], Maria Suzana Marques[1,2], Fernanda Piana Santos Lima de Oliveira[2]*, Daniela Marcia Rodrigues Caldeira[3‡], Geraldo Edson Souza Guerra Júnior[2‡], Ronilson Ferreira Freitas[1‡], Yaroslav Wladmir Lopes Popoff[2], Gustavo Veloso Rabelo[2‡], Mirna Rossi Barbosa-Medeiros[1‡], Luiza Augusta Rosa Rossi-Barbosa[1]

1 State University of Montes Claros, Montes Claros, Minas Gerais, Brazil, 2 Fipmoc University Center (UNIFIPMoc), Montes Claros, Minas Gerais, Brazil, 3 United College of Northern Minas, Montes Claros, Minas Gerais, Brazil

☯ These authors contributed equally to this work.
‡ These authors also contributed equally to this work.
* fernandapiana@gmail.com

## Abstract

This study aims to investigate the prevalence and factors associated with abdominal obesity in climacteric women assisted at Family Health Strategy units of the city of Montes Claros, State of Minas Gerais, Brazil. It is a cross-sectional analytical study. The women were selected by probabilistic sampling from August 2014 to August 2015. A questionnaire containing information referring to sociodemographic and economic characteristics, behavioral characteristics and clinical data was used. To estimate abdominal obesity, the measure of circumference $\geq$ 88 cm was considered. To analyze the association between abdominal obesity and the independent variables, a bivariate analysis was performed by means of Pearson's chi-square test (p$\leq$0.25). Subsequently, a multiple Poisson regression analysis with robust variance was performed, through which prevalence ratios with level of significance of 5% (p<0.05) were obtained. A total of 805 women were evaluated, aged 40 to 65 years, and the prevalence of women with abdominal obesity was 62.4%. The mean and median of abdominal circumference were 93.0 cm. The associated variables were being sedentary (PR = 1.44) or irregularly active (PR = 1.39), presenting altered total cholesterol (PR = 1.21), and being hypertensive (PR = 1.31). The abdominal obesity in climacteric women was associated with physical inactivity, total cholesterol and arterial hypertension. The measurement of abdominal circumference must be valued and adopted in the routine of professionals who work in Primary Care.

**Data Availability Statement:** All relevant data are within the manuscript and its Supporting Information files.

**Funding:** The author(s) received no specific funding for this work.

**Competing interests:** The authors have declared that no competing interests exist.

## Introduction

The climacteric is a natural phase of the woman's life that comprises the transition between the reproductive and the non-reproductive period [1]. Biologically, it is understood as a set of changes in the ovarian structure and function, with a decrease in the production of steroid hormones [2]. This important life stage starts when the woman is approximately 40 years old and lasts until the age of 60–65 years. It is characterized by hormonal changes and functional, morphological, psychogenic and physical modifications [3]. It is believed that, of life's several phases, the climacteric is the one that causes the greatest impact on the woman's health [4].

Among different changes that affect the health of the climacteric woman, obesity occupies a prominent position. This pathology has a complex and multifactorial etiology with an epidemic character [5] and is considered a severe public health problem worldwide [3]. In Brazil, data from the Surveillance System of Risk and Protection Factors for Chronic Diseases by Telephone Survey (VIGITEL) of the year of 2017 showed that overweight was present in 50.5% of the women, increasing as age increases up to 64 years [6]. Moreover, a study conducted with the climacteric population found that more than 35.0% of these women were diagnosed with obesity [7].

During the climacteric stage, fat accumulation is redistributed from peripheral locations to intra-abdominal deposits, resulting in abdominal obesity. This has been one of the most worrying climacteric symptoms among women [8], and can be related to genetic, exogenous and mainly hormonal factors. In women, estrogen is the hormone responsible for fat accumulation in the subcutaneous tissue, particularly in the gluteal and femoral regions, while the androgens are responsible for abdominal fat accumulation [9]. However, in this phase, estrogen levels decrease, causing hyperandrogenemia and resulting in fat redistribution from the gynoid to the android (abdominal) position [9–11].

Abdominal obesity has been one of the main public health problems and has drawn researchers' attention, not only because it affects a considerable number of women, but also due to its strong correlation with an increase in morbidity and mortality risks [12]. Thus, researching into abdominal obesity has become relevant, as the conditions of women, mainly those in the climacteric phase, represent a challenge to Primary Care due to its low capacity for meeting these women's real needs. In this context, the present study aimed to investigate the prevalence and factors associated with abdominal obesity in climacteric women assisted at Family Health Strategy units of the city of Montes Claros—State of Minas Gerais.

## Methods

This is a cross-sectional, analytical study that is part of a larger project called "Health problems in climacteric women: An epidemiological study". It was developed at the Family Health Strategy units of the city of Montes Claros, State of Minas Gerais, Brazil. The Family Health Strategy is a project of Brazil's Ministry of Health to organize Primary Care and the Family Health units, with the aim of providing quality of life for the Brazilian population and intervening in factors that put health at risk [13].

The aforementioned project has a robust database, collected in 2015, with the participation of several researchers in the area, with two published articles [14, 15], both cross-sectional studies. The first evaluated the quality of life and associated factors of climacteric women [14]. The second investigated the association between health conditions and overweight in climacteric women, and in addition to the outcome variable, overweight and obesity, other variables such as sociodemographic, reproductive, clinical, eating and behavioural factors were evaluated [15]. In this study, the purpose was to verify the variables associated with waist circumference because we believe it is relevant to adopt it as a routine measure in Primary Care for

climacteric women, with evidences between this and physical inactivity, the increase in the total levels of cholesterol and hypertension.

The final sample of the present study differs from others in the same population group previously published [14, 15], since women who were not attended to measure waist circumference, a dependent variable in this study, were excluded from the sample. Those who did not answer the questionnaire about food were also excluded, as it is believed to be essential to carry out the association with the dependent variable.

The target population was composed of 30,018 climacteric women enrolled in 73 healthcare units. Pregnant, postpartum or bedridden women were excluded, and women who did not attend the data collection session after three attempts were considered losses. For sample size determination, the formula to estimate prevalence in cross-sectional studies was used [16]. To perform the calculation, a confidence level of 95% was adopted, with precision of 5% and a 50% prevalence of abdominal obesity. An adjustment for design effect was performed and a deff of 2.0 was adopted.

The sample was selected by probabilistic two-stage cluster sampling. In the first stage, the method of probability proportional to size (PPS) was used, and 20 Family Health Strategy units were selected in urban and rural areas. In the second stage, 48 women were drawn by simple random sampling (SRS) in each selected unit. To incorporate the structure of the complex sampling design into the statistical analysis, each woman was associated to a weight (w), which corresponded to the inverse of that woman's probability of being included in the sample (f) [17]. Women aged between 40 and 65 years who were physically able to answer the questionnaires were considered eligible to participate in the study. The calculations determined a sample size of at least 760 women.

A pilot study was carried out at one Family Health Strategy unit, with women belonging to the studied age group who were not selected for the final sample. This enabled to test the questionnaire and the interviewers' performance in practice. After this stage, field research started. It was not necessary to make adjustments to the data collection instrument. Subsequently, the women were invited to participate in the research on a previously established date. Data collection was performed at the Family Health Strategy units. The final sample that composed our study, considering the missing data and without interfering with the required minimum sample size, contained 805 climacteric women. All the participants signed a consent document. They underwent anthropometric measurements and laboratory tests (after a 12-hour fast).

Abdominal obesity, assessed by the measurement of abdominal circumference, was considered the dependent variable and the cut-off point $\geq$ 88 cm was used, in accordance with the National Cholesterol Education Program's Adult Treatment Panel III (NCEP/ATP-III) [18], which has been employed by other studies [11, 19, 20] with similar population groups. Abdominal circumference was measured with a flexible, inelastic tape of the TBW$^{®}$ brand that has a graduation interval of 0.1 cm. Abdominal circumference was measured directly on the skin, at the midpoint between the last rib and the iliac crest [21]. During the measurement, the woman remained in orthostatic position, with arms along the body, abdomen relaxed, looking at a fixed point in front of her [18]. This variable was dichotomized (obese and not obese) and the measure $\geq$ 88 cm was considered abdominal obesity.

The independent variables were the sociodemographic and economic data (age group, marital status, level of schooling, number of pregnancies, family income), behavioral data (intake of meat with fat, intake of chicken skin, salt in the food, fruit consumption, soft drink consumption, smoking, alcohol consumption, and physical activity level), and clinical data (climacteric stages, climacteric symptoms, depressive symptoms, anxiety symptoms, sleep quality, hormone therapy, glucose, triglycerides, total cholesterol, High Density Lipoproteins (HDL), and blood pressure).

Age group (40 to 45; 46 to 51; 52 to 65) was measured by the investigation of the woman's age: her birth month and year were asked and confirmed in an identity document. Marital status was dichotomized as with or without partner [22]. Level of schooling was investigated by asking the woman about the highest education level she achieved. The correspondence was performed in such a way that each grade corresponded to one year of study [22], categorized into three classes: primary education, secondary education and higher education. The reproductive variable was investigated by number of pregnancies and was categorized as $\leq$ 1; 2; and $\geq$ 3 [18].

Gross monthly family income was classified according to the amount of minimum salaries the family earned per month (when the research was carried out, one minimum salary equaled R$ 724.00—seven hundred and twenty-four reais), and the answers were grouped into two categories: > 1 minimum salary and $\leq$ 1 minimum salary.

The questions about eating habits (intake of meat with fat, of chicken skin, salt in the food, fruit consumption, soft drinks) were based on the VIGITEL questionnaire [6].

Intake of meat with fat and chicken with skin had five possibilities of answers and were dichotomized as no and yes. The women who answered "Always removes the visible excess"; "Does not eat red meat with too much fat nor chicken with skin"; "Does not eat red meat nor chicken skin" were considered non-consumers.

About consumption of salt in the food, there were three possible answers that were dichotomized as no and yes. The affirmative was considered "I put salt almost always, even without tasting the food". As for fruit consumption, the question referred to daily consumption and was dichotomized as: $\geq$ 3 fruit servings a day and $\leq$ 2 fruit servings a day. Concerning consumption of soft drinks, it was classified into: < 3 times a week and $\geq$ 3 times a week.

Smoking and alcohol consumption were also based on VIGITEL [6], and the women who answered they smoke cigars or similar products were considered smokers. Alcohol consumption was considered the intake of four or more doses in one single occasion, in the 30 previous days. One dose is equivalent to one can of beer, one glass of wine or one shot of distilled spirits. Both variables were dichotomized as no and yes.

Physical activity level was measured by means of the instrument International Physical Activity Questionnaire (IPAQ), short version validated in Brazil [23], which classifies the person as very active, active, irregularly active and sedentary. Very active is the person who practices vigorous physical activity more than five days a week, 30 minutes per session. Active is the one who complies with the recommendation of practicing vigorous physical activity more than three days a week, in sessions that last more than 20 minutes. Irregularly active refers to the person who practices physical activity during at least ten minutes per week, which is insufficient to categorize them as active. Sedentary is the person who does not practice physical activities during at least ten continuous minutes during the week. The variable physical activity was categorized as active, irregularly active and sedentary women.

The climacteric stages were classified in the following way: premenopausal women, when their menstrual cycle was regular (28 to 28 days, 29 to 29 days), perimenopausal women, when their menstrual cycle was irregular, varying from 2 to 11 months, and postmenopausal women, when their menstrual cycle had been interrupted for more than 12 months [24].

The climacteric symptoms were described based on the Kupperman Index [25], in which the following symptoms are graduated as mild, moderate and intense: vasomotor, paresthesia, insomnia, nervousness, sadness, weakness, arthralgia/myalgia, headache, palpitations, tingling. Subsequently, this variable was dichotomized as mild and moderate/intense. Regarding hormone therapy, it was investigated whether the women were undergoing it or not.

Anxiety and depression symptoms were assessed by the Beck Anxiety Inventory (BAI) [26] and by the Beck Depression Inventory (BDI) [27], translated and validated for Brazil [28, 29]. The BAI is composed of 21 questions, each with four possibilities of answers, about how the individual felt in the previous week, expressed in common anxiety symptoms. The maximum score is 63 points and the categories are: score 0–10, minimal anxiety; 11–15, mild anxiety; 20–30, moderate anxiety; 31–63, severe anxiety [28]. The symptoms were dichotomized as minimal/mild and moderate/severe.

The BDI is constituted of 21 items that include symptoms and attitudes classified into four degrees of intensity. Each item presents four options and it is possible to have more than one answer in each question, but only the alternative with the highest value is considered. Total scores can vary from 0 to 63, suggesting the following degree of severity: 0–9, normal; 10–15, mild depression; 16–23, moderate depression; and 24 or more, severe depression [29]. The variable was categorized as normal, mild and moderate/severe.

To assess sleep quality, the Pittsburgh Sleep Quality Index, developed by Buysse *et al*. [30] and translated and validated for Brazilian Portuguese (PSQI-BR) [31], was used. The instrument assesses sleep quality in the four previous weeks and categorizes it as "well-preserved" and "impaired". It is composed of 19 questions that are analyzed by seven component scores: (1) subjective sleep quality; (2) sleep latency; (3) sleep duration; (4) habitual sleep efficiency; (5) sleep disturbances; (6) use of sleeping medication; (7) sleepiness or daytime dysfunction. The sum of these seven components yields one global score that varies from 0 to 20, where 0–4 indicate a well-preserved sleep quality and 5–20 indicate an impaired sleep quality.

Blood samples were collected for the tests of fasting blood glucose, triglycerides, total cholesterol, and HDL. Glucose results ≥100mg/dL were considered altered and <100mg/dL, normal. Triglycerides ≥150 mg/dl was considered altered and <150 mg/dl, normal. As for total cholesterol levels, results ≤ 190 mg/dL were considered desirable and > 190 mg/dL were considered altered. For HDL cholesterol, results > 40 mg/dL were considered normal and ≤ 40 mg/dL were considered low [32].

Blood pressure was measured using a calibrated aneroid sphygmomanometer of the OMRON® brand placed on the proximal region of the left upper limb above the cubital fossa, with the patient seated, after a five-minute rest. Blood pressure was measured twice, with a one-minute interval between measures, and the mean of the results was calculated. Hypertensive women were considered those with Systolic Blood Pressure (SBP) ≥ 140 mmHg and/or Diastolic Blood Pressure (DBP) ≥ 90 mmHg [33].

Data were tabulated with the use of the Predictive Analytics Software (PASW) 20.0. Initially, descriptive analyses of all the variables were performed to determine their distribution and frequencies. To analyze the association between abdominal obesity (dependent variable) and the independent variables, a bivariate analysis was performed by means of Pearson's chi-square test. Considering that Poisson regression with robust variance provides correct estimates and is a better alternative for the analysis of cross-sectional studies with binary outcomes [34], crude prevalence ratios (PR) were calculated, with their respective confidence intervals of 95%. The variables that proved to be associated up to the level of 25% (p≤0.25) were selected for multiple Poisson regression analysis with robust variance, and adjusted prevalence ratios and their respective 95% confidence intervals (95% CI) were obtained. For the final model, the 5% level of significance (p<0.05) was adopted.

As this study involved human beings, it was submitted to, appraised and approved by the Research Ethics Committee of the higher education institution *Faculdades Integradas Pitágoras de Montes Claros*, under opinion no. 817.666 (CAAE 36495714.0.0000.51).

**Table 1. Distribution of absolute and relative frequencies and bivariate analysis between abdominal obesity and socio-demographic and economic variables in climacteric women.**

| SOCIODEMOGRAPHIC AND ECONOMIC DATA | | | Abdominal obesity | | |
|---|---|---|---|---|---|
| | | | Not obese < 88cm | Obese ≥ 88cm | *p-value* |
| Variables | n | % | % | % | $(x^2)$ |
| Age Group | | | | | |
| 40 to 45 years | 221 | 27.5 | 46.2 | 53.8 | 0.001 |
| 46 to 51 years | 220 | 27.3 | 40.5 | 59.5 | |
| 52 to 65 years | 364 | 45.2 | 30.8 | 69.2 | |
| Marital Status | | | | | |
| With partner | 518 | 64.5 | 39.2 | 60.8 | 0.251 |
| Without partner | 285 | 35.5 | 35.1 | 64.9 | |
| Level of Schooling | | | | | |
| Higher education | 42,2 | 5.3 | 50.0 | 50.0 | 0.011 |
| Secondary education | 227 | 28.2 | 42.7 | 57.3 | |
| Primary education | 452 | 56.1 | 33.2 | 66.8 | |
| Number of pregnancies | | | | | |
| ≤ 1 | 99 | 12.3 | 42.4 | 57.6 | 0.462 |
| 2 | 164 | 20.4 | 34.8 | 65.2 | |
| ≥ 3 | 542 | 67.3 | 37.6 | 62.4 | |
| Family income | | | | | |
| > 1 minimum salary | 457 | 56.8 | 39.4 | 60.6 | 0.241 |
| ≤ 1 minimum salary | 348 | 43.2 | 35.3 | 64.7 | |

* Minimum salary at the time, R$ 724.00 (seven hundred and twenty-four reais)

## Results

Overall, 805 women participated in the study, aged 40 to 65 years—mean: 50.8 years (SD±6.98); median: 50 years. The prevalence of women with abdominal obesity was 62.4%. The mean and median of abdominal circumference were 93.0 cm.

Tables 1, 2 and 3 present the absolute and relative frequencies, as well as the bivariate analyses between abdominal obesity and the investigated independent variables. The independent variables selected for the multiple model were: age group (p = 0.001), level of schooling (p = 0.010), family income (p = 0.241), intake of meat with fat (p = 0.192), fruit consumption (p = 0.217), physical activity level (p<0.001), climacteric stage (p<0.001), climacteric symptoms (p = 0.253), hormone therapy (p = 0.166), glucose (p = 0.121), triglycerides (p = 0.055), total cholesterol (0.001), and arterial hypertension (p<0.001).

When we performed the adjusted multiple model, we found that the women who were irregularly active and sedentary, those with altered cholesterol levels, and the hypertensive women remained associated with abdominal obesity. The prevalence ratios with their respective confidence intervals are presented in Table 4.

## Discussion

This study revealed a high prevalence of abdominal obesity among climacteric women, as approximately two thirds of the participants presented this health problem. Such high occurrence was associated with irregularly active and sedentary women, women with altered cholesterol levels, and hypertensive women.

**Table 2. Distribution of absolute and relative frequencies and bivariate analysis between abdominal obesity and behavioral variables in climacteric women.**

| BEHAVIORAL DATA | | | Abdominal obesity | | |
|---|---|---|---|---|---|
| | | | Not obese < 88cm | Obese ≥ 88cm | p-value |
| Variables | n | % | % | % | ($x^2$) |
| Intake of meat with fat | | | | | |
| No | 665 | 83.3 | 36.8 | 63.2 | 0.192 |
| Yes | 133 | 16.7 | 42.9 | 57.1 | |
| Intake of chicken skin | | | | | |
| No | 727 | 91.1 | 38.2 | 61.8 | 0.616 |
| Yes | 71 | 8.9 | 35.2 | 64.8 | |
| Salt in food | | | | | |
| No | 789 | 98.1 | 37.6 | 62.4 | 0.852 |
| Yes | 15 | 1.9 | 40.0 | 60.0 | |
| Fruit consumption | | | | | |
| ≥ 3 servings per day | 284 | 35.3 | 40.5 | 59.5 | 0.217 |
| ≤ 2 servings per day | 521 | 64.7 | 36.1 | 63.9 | |
| Soft drink consumption | | | | | |
| < 3 times a week | 714 | 88.9 | 38.0 | 62.0 | 0.714 |
| ≥ 3 times a week | 89 | 11.1 | 36.0 | 64.0 | |
| Smoking | | | | | |
| No | 694 | 86.9 | 38.2 | 61.8 | 0.442 |
| Yes | 105 | 13.1 | 34.3 | 65.7 | |
| Alcohol consumption | | | | | |
| No | 747 | 92.8 | 38.2 | 61.8 | 0.281 |
| Yes | 58 | 7.2 | 31.0 | 69.0 | |
| Physical activity level | | | | | |
| Very active | - | - | - | - | |
| Active | 104 | 12.9 | 55.8 | 44.2 | < 0.001 |
| Irregularly active | 449 | 55.8 | 34.3 | 65.7 | |
| Sedentary | 252 | 31.3 | 36.1 | 63.9 | |

Prevalence of abdominal obesity was similar to the values found in other studies carried out in Brazil [11, 19, 20] that used the same cut-off point of 88 cm. One of the studies was conducted with 456 postmenopausal women aged 45–69 years [19], another study involved climacteric women aged 40–65 years assisted at private gynecology offices in the State of Minas Gerais [11], and the third study was carried out with 201 women aged 44–65 years assisted at the Central Outpatient Clinic of the University of Caxias do Sul [20].

In articles in which the cut-off point for waist circumference was 80 cm, a similar prevalence was found among pre- and postmenopausal women [35], and the prevalence found in studies conducted in Maringá (State of Paraná) [36] and Rio de Janeiro [37] was lower.

In the Brazilian literature, in addition to different cut-off points for the assessment of abdominal adiposity, abdominal circumference and waist circumference are measured in distinct ways, which can generate incoherence in the interpretation of results [38]. The most used definition, which was employed in the present study, determines that abdominal circumference is located between the last ribs and the iliac crest, in its largest perimeter, and can coincide with the umbilical scar or not. Waist circumference, in turn, is measured at the point that has the smallest perimeter of the region [21]. This information is relevant because different cut-off points and measurement locations generate equivocal prevalence values, hindering the design

**Table 3. Distribution of absolute and relative frequencies and bivariate analysis between abdominal obesity and clinical variables in climacteric women.**

| CLINICAL DATA | | | Abdominal obesity | | |
|---|---|---|---|---|---|
| | | | Not obese < 88cm | Obese ≥ 88cm | p-value |
| Variables | n | % | % | % | $(x^2)$ |
| Climacteric Stages | | | | | |
| Premenopause | 219 | 27.3 | 46.1 | 53.9 | 0.001 |
| Perimenopause | 220 | 27.4 | 40.5 | 59.5 | |
| Postmenopause | 364 | 45.3 | 30.8 | 69.2 | |
| Climacteric Symptoms | | | | | |
| Mild | 491 | 61.1 | 39.3 | 60.7 | 0.235 |
| Moderate/Intense | 313 | 38.9 | 35.1 | 64.9 | |
| Hormone therapy | | | | | |
| No | 746 | 93.9 | 37.0 | 63.0 | 0.166 |
| Yes | 49 | 6.1 | 46.9 | 53.1 | |
| Depressive Symptoms | | | | | |
| Normal | 485 | 60.2 | 39.0 | 61.0 | 0.571 |
| Mild | 205 | 25.5 | 36.6 | 63.4 | |
| Moderate / Severe | 112 | 13.9 | 33.9 | 66.1 | |
| Anxiety Symptoms | | | | | |
| Minimal / Mild | 552 | 68.7 | 37.1 | 62.9 | 0.683 |
| Moderate / Severe | 251 | 31.3 | 38.6 | 61.4 | |
| Sleep Quality | | | | | |
| Well-preserved | 617 | 77 | 38.7 | 61.3 | 0.269 |
| Impaired | 184 | 23 | 34.2 | 65.8 | |
| Glucose | | | | | |
| Normal | 633 | 78.6 | 39.0 | 61.0 | 0.121 |
| Altered | 172 | 21.4 | 32.6 | 67.4 | |
| Triglycerides | | | | | |
| Normal | 356 | 53 | 41.0 | 59.0 | 0.055 |
| Altered | 402 | 47 | 34.3 | 65.7 | |
| Total cholesterol | | | | | |
| Desirable | 302 | 39.5 | 42.7 | 57.3 | 0.001 |
| Altered | 461 | 60.5 | 29.5 | 70.5 | |
| HDL cholesterol | | | | | |
| Normal | 302 | 39.6 | 40.1 | 59.9 | 0.313 |
| Low | 461 | 60.4 | 36.4 | 63.6 | |
| Blood pressure | | | | | |
| Not hypertensive | 511 | 63.9 | 45.6 | 54.4 | < 0.001 |
| Hypertensive | 289 | 36.1 | 23.2 | 76.8 | |

of strategies for improved prevention. It is important to standardize the cut-off point and its measurement for the investigated population.

In relation to the mean of the abdominal circumference, a study [39] carried out with post-menopausal women found a mean of 94.8 cm. In another study [40] involving women aged between 40 and 65 years, the mean was 95.7 cm. These results are similar to the ones found in our study and show that the climacteric brings changes in body composition. Abdominal obesity is an independent parameter even among people with normal BMI [41]; therefore, the measurement of abdominal circumference should be routinely performed by Primary Care professionals in climacteric women. When this measure increases, it becomes an indicator for

**Table 4. Factors associated with abdominal obesity in climacteric women.**

| Variables | PR (95%CI)* adjusted |
|---|---|
| Physical activity level | |
| Active | 1 |
| Irregularly active | 1.39 (1.09–1.78) |
| Sedentary | 1.44 (1.13–1.82) |
| Cholesterol | |
| Normal | 1 |
| Altered | 1.21 (1.07–1.37) |
| Blood pressure | |
| Not hypertensive | 1 |
| Hypertensive | 1.31 (1.18–1.47) |

(*) PR: Prevalence ratio; 95%CI: Confidence Interval of 95%

the development of metabolic changes which, associated, can increase the risk of cardiovascular problems.

In our study, abdominal obesity was associated with irregularly active and sedentary women. A research carried out in the United States from 1988 to 2010 found that the association between abdominal obesity and level of physical activity is significant [41]. Physical activity protects against cardiovascular problems and is a preventive, non-pharmacological method; therefore, it must be a routine practice [37, 41].

A study that compared results before and after a standard twelve-week exercise program revealed a significant reduction in abdominal circumference [39]. Another study was carried out with postmenopausal women in the threshold for overweight. One group was submitted to a physical training program and the control group was authorized to continue with its normal level of physical activity. After one year, a reduction in waist circumference was found in both groups, and it was significant in the group that underwent the training program [41].

A divergent result was found in a study with postmenopausal Nigerian women, whose objective was to investigate the association between level of physical activity, general obesity and abdominal obesity. The researchers did not find a significant association of these variables after a logistic regression analysis [42].

A prospective Danish research with 26,625 middle-aged healthy individuals aimed to investigate the association between waist circumference alterations and mortality. It concluded that this association was positive, as individuals can benefit from a weight loss that selectively reduces abdominal fat, but if the weight loss reduces the lean body mass, this can cause harmful effects on health [43]. This result shows the importance of physical activity so that the individual does not lose lean body mass when he/she loses weight.

Total cholesterol was another variable associated with abdominal obesity. A similar result was found in a study conducted with obese climacteric women assisted at Family Health Strategy units, which identified high total cholesterol levels [44]. In a study conducted in the State of Minas Gerais with postmenopausal women, it was possible to see that the majority presented high levels of total cholesterol [45]. A research carried out in the State of Rio Grande do Sul found that postmenopausal women, in addition to presenting higher values of abdominal circumference, also presented higher values of total cholesterol when compared to premenopausal women [46].

The association between dyslipidemia and obesity is frequent, mainly in older individuals. The increase in cholesterol levels is explained by metabolic complications, as there is a

deregulation of the lipolysis process that results in a greater release of fatty acids and glycerol. Adipose tissue is the body's largest cholesterol reservoir [47].

Hypertension was associated with abdominal obesity and this result was also found in a cross-sectional study with rural Chinese women aged 35 to 65 years and older [48]. A cohort study [49] carried out in China for 22 years with 12,907 participants showed that abdominal circumference and body mass index predicted the development of hypertension. As these measures are simple, effective and widely applicable, the authors recommend that they should be used as predictors of hypertension in public health strategies [49]. According to them, measuring abdominal circumference is necessary because not only the amount of fat, but also the location of specific fat deposits is important for the development of hypertension [49, 50].

In Brazilian studies [37, 51], hypertension has also been associated with abdominal obesity. The study conducted in Rio de Janeiro [37] that found a high prevalence of abdominal obesity also showed a higher chance of having arterial hypertension. A study carried out with women assisted at a Family Health Strategy unit located in the State of São Paulo, in which the cut-off point was 88 cm, showed that a large abdominal circumference increased the prevalence of hypertension [51]. A study with hypertensive individuals showed that they presented higher values of anthropometric measures and that abdominal circumference can predict hypertension and be a useful screening tool [52].

The mechanisms of hypertension in obesity are various, but this association has not been totally explained yet. It is known that blood pressure increases with weight gain and with visceral deposition [49, 53].

The present study has a cross-sectional design that does not allow to establish cause-and-effect relationships. Another limitation is that it is based on self-reports to assess behavioral aspects and some issues of clinical aspects, like climacteric stages and symptoms, depressive and anxiety symptoms, and sleep quality. However, the relevance of the results must be highlighted. In addition to having a large selected sample, representative of the population, it was possible to see that the associated variables suggest that the measurement of abdominal circumference must be considered important and incorporated by Primary Care professionals. After all, it is easy, fast and has no cost. In addition, it is important to emphasize that physical activity, the monitoring of blood pressure levels and of cholesterol levels must be encouraged during clinical practice.

## Conclusion

In the present study with climacteric women, the prevalence of abdominal obesity was high and was associated with physical inactivity, total cholesterol levels and hypertension. In light of these results, we believe that the measurement of abdominal circumference must be valued and adopted as a routine procedure by Primary Care professionals.

Further research, mainly longitudinal studies, can contribute to clarify this marker for climacteric women's health.

## Supporting information

**S1 Data.**
(SAV)

## Author Contributions

**Conceptualization:** Wiviane da Costa Pimenta, Josiane Santos Brant Rocha, Luiza Augusta Rosa Rossi-Barbosa.

**Data curation:** Josiane Santos Brant Rocha, Antônio Prates Caldeira, Daniela Araújo Veloso Popoff.

**Formal analysis:** Josiane Santos Brant Rocha, Antônio Prates Caldeira, Daniela Araújo Veloso Popoff, Viviane Maia Santos, Joyce Elen Murça de Souza, Maria Suzana Marques, Fernanda Piana Santos Lima de Oliveira, Luiza Augusta Rosa Rossi-Barbosa.

**Investigation:** Wiviane da Costa Pimenta, Josiane Santos Brant Rocha, Daniela Marcia Rodrigues Caldeira, Geraldo Edson Souza Guerra Júnior, Ronilson Ferreira Freitas, Yaroslav Wladmir Lopes Popoff, Gustavo Veloso Rabelo, Mirna Rossi Barbosa-Medeiros, Luiza Augusta Rosa Rossi-Barbosa.

**Methodology:** Wiviane da Costa Pimenta, Josiane Santos Brant Rocha, Antônio Prates Caldeira, Daniela Araújo Veloso Popoff, Viviane Maia Santos, Joyce Elen Murça de Souza, Maria Suzana Marques, Fernanda Piana Santos Lima de Oliveira, Luiza Augusta Rosa Rossi-Barbosa.

**Project administration:** Josiane Santos Brant Rocha, Luiza Augusta Rosa Rossi-Barbosa.

**Resources:** Wiviane da Costa Pimenta, Josiane Santos Brant Rocha, Viviane Maia Santos, Joyce Elen Murça de Souza, Maria Suzana Marques, Fernanda Piana Santos Lima de Oliveira, Luiza Augusta Rosa Rossi-Barbosa.

**Software:** Wiviane da Costa Pimenta, Josiane Santos Brant Rocha, Antônio Prates Caldeira, Daniela Araújo Veloso Popoff, Viviane Maia Santos, Joyce Elen Murça de Souza, Maria Suzana Marques, Fernanda Piana Santos Lima de Oliveira, Luiza Augusta Rosa Rossi-Barbosa.

**Supervision:** Josiane Santos Brant Rocha, Luiza Augusta Rosa Rossi-Barbosa.

**Validation:** Josiane Santos Brant Rocha, Antônio Prates Caldeira, Daniela Araújo Veloso Popoff, Luiza Augusta Rosa Rossi-Barbosa.

**Visualization:** Josiane Santos Brant Rocha, Viviane Maia Santos, Joyce Elen Murça de Souza, Maria Suzana Marques, Fernanda Piana Santos Lima de Oliveira, Luiza Augusta Rosa Rossi-Barbosa.

**Writing – original draft:** Wiviane da Costa Pimenta, Josiane Santos Brant Rocha, Luiza Augusta Rosa Rossi-Barbosa.

**Writing – review & editing:** Josiane Santos Brant Rocha, Antônio Prates Caldeira, Daniela Araújo Veloso Popoff, Viviane Maia Santos, Joyce Elen Murça de Souza, Maria Suzana Marques, Fernanda Piana Santos Lima de Oliveira, Daniela Marcia Rodrigues Caldeira, Geraldo Edson Souza Guerra Júnior, Ronilson Ferreira Freitas, Yaroslav Wladmir Lopes Popoff, Gustavo Veloso Rabelo, Mirna Rossi Barbosa-Medeiros, Luiza Augusta Rosa Rossi-Barbosa.

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
