## [Decision Letter · Decision Letter 0]

17 Mar 2020

PONE-D-20-04707

ABDOMINAL OBESITY ASSOCIATED WITH SOCIODEMOGRAPHIC, BEHAVIORAL AND CLINICAL DATA IN CLIMACTERIC WOMEN ASSISTED IN PRIMARY HEALTH CARE

PLOS ONE

Dear Dr. Piana Santos Lima de Oliveira,

Thank you for submitting your manuscript to PLOS ONE. After careful consideration, we feel that it has merit but does not fully meet PLOS ONE’s publication criteria as it currently stands. Therefore, we invite you to submit a revised version of the manuscript that addresses the points raised during the review process.

SPECIFIC ACADEMIC EDITOR COMMENTS: Your manuscript was handled by an expert reviewer in the field. Although interest was found in your study, there were several major comments that arose during the review. For instance, it does not seem as if a thorough review of the literature was included to rationalize the experimental design; several vague comments require clarification; and there are suggestions to strengthen the data presentation.

We would appreciate receiving your revised manuscript by May 01 2020 11:59PM. To enhance the reproducibility of your results, we recommend that if applicable you deposit your laboratory protocols in protocols.io, where a protocol can be assigned its own identifier (DOI) such that it can be cited independently in the future. For instructions see: http://journals.plos.org/plosone/s/submission-guidelines#loc-laboratory-protocols

We look forward to receiving your revised manuscript.

Kind regards,

Frank T. Spradley

Academic Editor

PLOS ONE

3. Thank you for including your ethics statement:

"The participants agreed to respond to this research voluntarily by signing the Free and Informed Consent Form. The project was submitted, analyzed and approved for execution by the Ethics and Research Committee, nº 817.666 (CAAE 36495714.0.0000.51)."

i) Please amend your current ethics statement to include the full name of the ethics committee/institutional review board(s) that approved your specific study.

ii) Once you have amended this statement in the Methods section of the manuscript, please add the same text to the “Ethics Statement” field of the submission form (via “Edit Submission”).

4. We note that your recently published article (Marques, Maria Suzana, et al. "Health conditions associated with overweight in climacteric women." PloS one 14.12 (2019).), is related to the present study, as the same sample population was considered, and the same scale adopted. To meet PLOS ONE criteria on related manuscripts (http://journals.plos.org/plosone/s/submission-guidelines#loc-related-manuscripts), we would ask that the previous analysis is adequately mentioned and cited (not only in the Abstract) in the present submission, and the rationale of these separate analyses is clearly discussed; moreover, please discuss why the number of participants included in the two studies is slightly different.

5. Please include additional information regarding the survey or questionnaire used in the study and ensure that you have provided sufficient details that others could replicate the analyses. For instance, if you developed a questionnaire as part of this study and it is not under a copyright more restrictive than CC-BY, please include a copy, in both the original language and English, as Supporting Information. Moreover, please include more details on how the questionnaire was pre-tested, and whether it was validated.

Reviewers' comments:

Reviewer's Responses to Questions

**Comments to the Author**

1. Is the manuscript technically sound, and do the data support the conclusions?

Reviewer #1: No

2. Has the statistical analysis been performed appropriately and rigorously? 

Reviewer #1: No

3. Have the authors made all data underlying the findings in their manuscript fully available?

Reviewer #1: Yes

4. Is the manuscript presented in an intelligible fashion and written in standard English?

Reviewer #1: No

5. Review Comments to the Author

Reviewer #1: The literature review seems not have been fully explored in order to support the choice of variables included in study.

It is not clear if this study is part of a bigger research project. If yes, it should be mentioned and explained, specially about the strategies of recruitment and selection of participants, quality control protocols for anthopometric mesures and choice of the questionaires.

It should be explained that Family Health Strategy is a Brazilian government strategy for organizing the primary health care, and the Family Health Care Units where people receive health care. It should be informed the referent cathegory for each variable, it is not clear. The translation and validation of the Pittsburgh Sleep Quality Index was not mentioned. There are some part of text in portugues, please see lines 191-192, 296-297.

The tables should be revised. Please, see the absolute numbers and percentages of variables "tryglicedies" and "total cholesterol". The absolute numbers of cathegories "not obese" and "obese" could be excluded, lefting just the percentages. On the table 4 should be excluded p value since it already show confidence intervals.

Please, do not present percentages on the discussion section. Data discussion seems not to be deeply explored by authors, specially not regarding gender aspects. The transversal framework of study is a limitation that was not mentioned. Conclusion are not properly presented, this manuscript section should not show percentagem.

6. PLOS authors have the option to publish the peer review history of their article (what does this mean?). If published, this will include your full peer review and any attached files.

Reviewer #1: No

---

## [Author Response · Author response to Decision Letter 0]

7 Jun 2020

Dear Editor and Reviewers.

Attending the reviewers’ recommendations, we described below the modifications made in the manuscript.

RESPONSE: It was included

2) We note that your recently published article (Marques, Maria Suzana, et al. "Health conditions associated with overweight in climacteric women." PloS one 14.12 (2019).), is related to the present study, as the same sample population was considered, and the same scale adopted. To meet PLOS ONE criteria on related manuscripts (http://journals.plos.org/plosone/s/submission-guidelines#loc-related-manuscripts), we would ask that the previous analysis is adequately mentioned and cited (not only in the Abstract) in the present submission, and the rationale of these separate analyses is clearly discussed; moreover, please discuss why the number of participants included in the two studies is slightly different.

RESPONSE: This part of the methods section has been included for better understanding.

Regarding the different number in the final sample, the first reason was due to the classification of menopause, we used the one performed by the sample and not by age, therefore having seven people missing. The second reason referred to those women who did not answer the questionnaire about food and we believe it is essential for the dependent variable. But we think it is unnecessary to include this justification in the article once it has met the necessary number of participants according to the sample calculation.

3) Please include additional information regarding the survey or questionnaire used in the study and ensure that you have provided sufficient details that others could replicate the analyses. For instance, if you developed a questionnaire as part of this study and it is not under a copyright more restrictive than CC-BY, please include a copy, in both the original language and English, as Supporting Information. Moreover, please include more details on how the questionnaire was pre-tested, and whether it was validated.

RESPONSE: Additional information was included.

4) Please include captions for your Supporting Information files at the end of your manuscript, and update any in-text citations to match accordingly. Please see our Supporting Information guidelines for more information: http://journals.plos.org/plosone/s/supporting-information.

RESPONSE: It was included.

5) The literature review seems not have been fully explored in order to support the choice of variables included in study.

RESPONSE: The literature review was revised and supports the choice of variables included in the study. We would like to point out that the manuscript is part of a master's dissertation, where the literature review was evaluated by qualification and defense boards.

6) It is not clear if this study is part of a bigger research project. If yes, it should be mentioned and explained, specially about the strategies of recruitment and selection of participants, quality control protocols for anthopometric mesures and choice of the questionaires.

RESPONSE: We have added this information in the text.

7) It should be explained that Family Health Strategy is a Brazilian government strategy for organizing the primary health care, and the Family Health Care Units where people receive health care.

RESPONSE: It was explained.

8) It should be informed the referent cathegory for each variable, it is not clear. 

RESPONSE: The categories of the variables were better informed.

9) The translation and validation of the Pittsburgh Sleep Quality Index was not mentioned. There are some part of text in Portuguese, please see lines 191-192, 296-297.

RESPONSE: We included this information. We corrected the parts that were written in Portuguese”. All text has been revised to English. 

10) The tables should be revised. Please, see the absolute numbers and percentages of variables "tryglicedies" and "total cholesterol". The absolute numbers of cathegories "not obese" and "obese" could be excluded, lefting just the percentages. 

RESPONSE: The tables have been revised. We excluded the absolute numbers.

11) On the table 4 should be excluded p value since it already show confidence intervals.

RESPONSE: P value was excluded.

12) Please, do not present percentages on the discussion section. 

RESPONSE: We removed the percentages from the discussion section.

13) Data discussion seems not to be deeply explored by authors, specially not regarding gender aspects. 

RESPONSE: We verified that there is no way to discuss gender aspects. Male gender was not an analyzed variable.

14) The transversal framework of study is a limitation that was not mentioned. 

RESPONSE: We added this limitation.

15) Conclusion are not properly presented, this manuscript section should not show percentage.

RESPONSE: The percentage has been removed from conclusion section.

Sincerely,

The authors

Dear Editor and Reviewers.

Attending the reviewers’ recommendations, we described below the modifications made in the manuscript.

1) 1) Thank you for your resubmission. We note that the following request has not been addressed: We note that your recently published article (Marques, Maria Suzana, et al. "Health conditions associated with overweight in climacteric women." PloS one 14.12 (2019).), is related to the present study, as the same sample population was considered, and the same scale adopted. To meet PLOS ONE criteria on related manuscripts (http://journals.plos.org/plosone/s/submission-guidelines#loc-related-manuscripts), we would ask that the previous analysis is adequately mentioned and cited (not only in the Cover letter) in the present submission, and the rationale of these separate analyses is clearly discussed.

RESPONSE: We have added this information in the text. The text has been revised accordingly so that this information is described in more detail (line 88-99).

Sincerely,

The authors

---

## [Editor Report · Decision Letter 1]

2 Jul 2020

PONE-D-20-04707R1

Abdominal obesity and association with sociodemographic, behavioral and clinical data in climacteric women assisted in Primary Care

PLOS ONE

Dear Dr. Piana Santos Lima de Oliveira,

Thank you for submitting your manuscript to PLOS ONE. After careful consideration, we feel that it has merit but does not fully meet PLOS ONE’s publication criteria as it currently stands. Therefore, we invite you to submit a revised version of the manuscript that addresses the points raised during the review process.

SPECIFIC ACADEMIC EDITOR COMMENTS: We apologize for the delay in reviewing your revised manuscript. The previous reviewer had issues with COVID-19 and had to opt out of re-reviewing your manuscript. However, as academic editor, I have proofed the manuscript and related revisions. Although the majority of comments were addressed appropriately, I think it would be a good idea to better state somewhere within lines 88-99 what is novel about this study by including a directional hypothesis statement (to match one included in the abstract and introduction), as this seems to be a re-analysis of data collected from an already-published cohort.

We look forward to receiving your revised manuscript.

Kind regards,

Frank T. Spradley

Academic Editor

PLOS ONE

---

## [Author Response · Author response to Decision Letter 1]

22 Jul 2020

Dear Editor,

Attending the specific academic editor comments, we described below the modifications made in the manuscript.

1) Although the majority of comments were addressed appropriately, I think it would be a good idea to better state somewhere within lines 88-99 what is novel about this study by including a directional hypothesis statement (to match one included in the abstract and introduction), as this seems to be a re-analysis of data collected from an already-published cohort.

RESPONSE: We have added this information in the text. The text has been revised accordingly so that this information is described in more detail (line 94-98). When reviewing the text, some information was moved from paragraph to paragraph as noted in the lines (119-120) and (129-130). All changes are highlighted in red in the body of the text.

Sincerely,

The authors

---

## [Editor Report · Decision Letter 2]

24 Jul 2020

Abdominal obesity and association with sociodemographic, behavioral and clinical data in climacteric women assisted in Primary Care

PONE-D-20-04707R2

Dear Dr. Piana Santos Lima de Oliveira,

We’re pleased to inform you that your manuscript has been judged scientifically suitable for publication and will be formally accepted for publication once it meets all outstanding technical requirements.

Kind regards,

Frank T. Spradley

Academic Editor

PLOS ONE

---

## [Editor Report · Acceptance letter]

4 Aug 2020

PONE-D-20-04707R2 

Abdominal obesity and association with sociodemographic, behavioral and clinical data in climacteric women assisted in Primary Care 

Dear Dr. Piana Santos Lima de Oliveira:

I'm pleased to inform you that your manuscript has been deemed suitable for publication in PLOS ONE. Congratulations! Your manuscript is now with our production department. 

Kind regards, 

on behalf of

Dr. Frank T. Spradley 

Academic Editor

PLOS ONE